Testing methods to mitigate Caribbean yellow-band disease on Orbicella faveolata

http://orcid.org/0000-0001-8112-3552 Randall Carly J. 1 2 crandall2012@my.fit.edu
Whitcher Elizabeth M. 1
Code Tessa 3
Pollock Clayton 3
Lundgren Ian 3
Hillis-Starr Zandy 3
Muller Erinn M. 4 emuller@mote.org
1 Florida Institute of Technology , Melbourne, FL , USA
2 Australian Institute of Marine Science , Townsville, QLD , Australia
3 National Park Service , St. Croix, VI , USA
4 Mote Marine Laboratory , Sarasota, FL , USA
Rodriguez-Lanetty Mauricio
Electronic publication date: 2018 May 11
Publication date: 2018
Volume: 6
Electronic Location ID: e4800
Received 2017 Nov 21; Accepted 2018 Apr 28
Copyright: © 2018 Randall et al.
Copyright year: 2018
Copyright holder: Randall et al.
License: This is an open access article distributed under the terms of the Creative Commons Attribution License, which permits unrestricted use, distribution, reproduction and adaptation in any medium and for any purpose provided that it is properly attributed. For attribution, the original author(s), title, publication source (PeerJ) and either DOI or URL of the article must be cited.
License URL: https://creativecommons.org/licenses/by/4.0/

Keywords: Yellow-band disease, Orbicella faveolata, Coral disease, Transmission, Aspiration, Shade, Firebreak, Chisel, Caribbean, Coral

Funding: National Park Service PMIS #168473 Cabbadetus Foundation Philanthropic Educational Organization (P. E. O.) Scholar Award Funding was provided by the National Park Service PMIS grant #168473 and by the Cabbadetus Foundation. C. J. Randall was partially supported by a Philanthropic Educational Organization (P. E. O.) Scholar Award. There was no additional external funding received for this study. The funders had no role in study design, data collection and analysis, decision to publish, or preparation of the manuscript.

==============================
Outbreaks of coral diseases continue to reduce global coral populations. In the Caribbean, yellow band is a severe and wide-spread disease that commonly affects corals of the Orbicella spp. complex, significantly impeding coral reproduction, and hindering the natural recovery of Orbicella spp. populations. Caribbean yellow-band disease (CYBD) lesions may be severe, and often result in the complete loss of coral tissue. The slow spread of CYBD, however, provides an opportunity to test methods to mitigate the disease. Here we report the results of in situ experiments, conducted within Buck Island Reef National Monument in St. Croix, USVI, to test the effectiveness of three techniques to minimize disease impact on Orbicella faveolata: (1) shading, (2) aspirating, and (3) chiseling a “firebreak” to isolate the lesion. Neither shading nor aspirating the diseased tissue significantly reduced CYBD tissue loss. However, chiseling reduced the rate and amount of tissue lost by 31%. While 30–40% of the chiseled lesions appeared to be free of disease signs 12–16 months after treatment, success significantly and steadily declined over 23 months, indicating a possible lack of long-term viability of the technique. The results of this study demonstrate that creating a “firebreak” between diseased and healthy-appearing tissue slows the spread of the disease and may prolong the life of O. faveolata colonies. The firebreak method yielded the best results of all the techniques tested, and also required the least amount of effort and resources. However, we do not recommend that this treatment alone be used for long-term disease mitigation. Rather, we propose that modifications of this and other treatment options be sought. The results also highlight the need for extended monitoring of CYBD after any treatment, due to the slow but variable rate and pattern of tissue loss in this disease.

Introduction

Outbreaks of coral diseases have contributed to the significant decline of coral populations in the Caribbean over the past four decades (Gardner et al., 2003; Weil, Cróquer & Urreiztieta, 2009), and both chronic and acute diseases have, and continue to, hamper coral recovery and restoration efforts throughout the region (Green & Bruckner, 2000; Rinkevich, 2005; Young, Schopmeyer & Lirman, 2012). Because coral-disease outbreaks are expected to continue in the future under increasingly stressful environmental conditions (Harvell et al., 2002; Randall & Van Woesik, 2017), there is an urgent need to develop methods to treat and minimize disease progression.

Caribbean yellow-band disease (CYBD) is a widely distributed and chronic disease that is most commonly observed on the critical reef-building corals in the Orbicella spp. complex (Reeves, 1994; Santavy et al., 1999; Cervino et al., 2001; Gil-Agudelo et al., 2004; Weil, 2004; Bruckner & Bruckner, 2006; Carricart-Ganivet, Beltrán-Torres & Horta-Puga, 2011; Soto-Santiago & Weil, 2014). CYBD is considered to be one of the most severe and lethal coral diseases, as recovery has rarely been observed (but see Weil, Cróquer & Urreiztieta, 2009 and Soto-Santiago & Weil, 2014 for seasonal dynamics). This disease was first reported on Orbicella faveolata in the Florida Keys by Reeves (1994), and was subsequently described by Santavy et al. (1999) as yellow-blotch disease in the San Blas Islands of Caribbean Panama. The prevalence of CYBD fluctuates through time, often peaking seasonally with elevated seawater temperatures (Cervino et al., 2001; Cróquer & Weil, 2009; Weil, Cróquer & Urreiztieta, 2009; Soto-Santiago & Weil, 2014; Randall & Van Woesik, 2017). While it was seldom reported prior to the 2005 Caribbean coral-bleaching event, CYBD is now common on many Caribbean reefs, necessitating the urgent development of treatment methods.

In the late 1980s, the first method to treat and manage a coral disease was developed and tested during an outbreak of black-band disease in Looe Key National Marine Sanctuary (Hudson, 2000). Researchers used an underwater aspiration system to suction the cyanobacterial mat off of the coral skeleton, and then used modeling clay, such as Roma plastilina sensu Fisher et al. (2007), to seal the disease-tissue interface (Hudson, 2000). After three years, the reinfection rate was low (∼30%), but black band was still present in the sanctuary, and treatment efforts were slow and laborious. Since then, various additional efforts to control coral-disease outbreaks have been proposed, with mixed success. These efforts include: (i) the use of biological control techniques such as probiotics and phage therapy to manage bacterial pathogens (Efrony, Atad & Rosenberg, 2009; Teplitski & Ritchie, 2009), (ii) the use of epoxy to mechanically block progression of a tissue-loss disease on Acropora cervicornis (Miller et al., 2014), (iii) the application of chlorine embedded within epoxy to treat and mechanically block progression of black-band disease (Aeby et al., 2015) and (iv) the excision of healthy coral branch tips from diseased A. cervicornis colonies (Miller et al., 2014). In addition, the mechanical removal of disease vectors, such as the corallivore Coralliophila abbreviata, has been proposed (Gignoux-Wolfsohn, Marks & Vollmer, 2012) but has not been implemented in a large-scale in situ experiment. The success of past mitigation techniques has been variable and dependent on the host species, the targeted etiological agent, and the disease vectors. For many coral diseases however, etiological agents are unknown, impeding the development of focused treatment methods.

The etiology of CYBD is not fully understood, but it is thought to be an infectious disease caused by a consortium of Vibrio spp. bacterial pathogens that act primarily on the host’s symbiotic dinoflagellates (Cervino et al., 2001, 2004a, 2004b, 2008; Cróquer et al., 2012). Pinpointing a causative agent has proved challenging, as distinct pathogenic bacteria are not consistently and reproducibly present in diseased tissue (Cróquer et al., 2012; Closek et al., 2014; Kimes et al., 2010; Kimes et al., 2013). Furthermore, Koch’s postulates have not been definitively satisfied for this disease. However, studies clearly show that the bacterial communities on and in diseased-coral tissues are distinct from the bacterial communities on and in healthy-coral tissues, and known pathogens are usually present in the diseased tissue (Cróquer et al., 2012; Closek et al., 2014; Kimes et al., 2010, but see Kimes et al., 2013). Increased prevalence and rates of spread of CYBD lesions also have been associated with several environmental stressors, including anomalously high sea-surface temperatures (Cervino et al., 2004a, 2004b; Cróquer & Weil, 2009; Weil, Cróquer & Urreiztieta, 2009), and high nutrient concentrations (Bruno et al., 2003). In combination, these studies indicate that CYBD may be strongly influenced by the environment. It is also possible that CYBD is caused by endogenous and ubiquitous microbes (sensu Thrusfield, 2007) that cause disease when the coral host or the coral symbionts become compromised. Indeed, the proposed causative bacterial pathogens have been found within the natural healthy-coral microbiota (Cróquer et al., 2012), but whether this is always true remains unknown and requires further investigation.

Caribbean yellow-band disease significantly and severely reduces reproductive output in O. faveolata through several mechanisms, including: (i) the direct reduction of live tissue, (ii) the reduction in fecundity of affected polyps and polyps adjacent to the lesion margin, and (iii) fragmentation of large tissue patches resulting in more tissue “edges” that are non-reproductive, and fewer large and reproductive ramets (Weil, Cróquer & Urreiztieta, 2009). Recent evidence gathered through infrared spectroscopy indicates that energy-rich proteins, carbohydrates, and phospholipidic compounds are reduced in diseased and marginal tissue, presumably leaving less energy for reproduction (Guerra et al., 2014). In addition, evidence suggests that CYBD disproportionately affects the largest (>1 m2) colonies in a population. For example, on a fringing reef off Salar Island, Panama, 70% of the largest O. faveolata colonies had CYBD while only 10% of all colonies had the disease (Santavy et al., 1999). Large colonies contribute disproportionately more reproductive material to the population, and by affecting the most fecund individuals, CYBD further limits the recovery potential of the population.

Because CYBD is now a common and chronic disease, there is an urgent need to develop methods to treat or slow its progression. Therefore, the objectives of this research were twofold: (1) to evaluate the effectiveness of three mitigation techniques in reducing disease progression; and (2) to refine and quantify the long-term success rate of the most promising technique.

Materials and Methods

Site selection

Caribbean yellow-band disease is a chronic problem on Caribbean reefs, and has been present within the Buck Island Reef National Monument in St. Croix, US Virgin Islands since at least 2005 (Miller et al., 2009). Prior monitoring within the national park identified hotspots of disease on large O. faveolata colonies on the fore reef to the east of Buck Island, at 10–15 m depth (E. Muller & C. Randall, 2014, unpublished data). Therefore, mitigation efforts were tested on colonies in this area with classic signs of the disease, defined as the presence of a gradient of tissue discoloration with apparently normal, thick, and pigmented tissue grading into thinner pale yellow tissue at the dying margin, which is usually smooth with an indistinct edge (Figure 1; sensu Work & Aeby, 2006). The algae biofouling community that colonizes the denuded skeleton is visible adjacent to the dying margin, rarely leaving a stark white skeleton. Tissue loss proceeds slowly as the lesion expands, either outwardly from centrally located lesions, or inwardly from peripherally located lesions (see Work & Aeby, 2006 for descriptions of terminology). We note that this project was conducted under permit BUIS-2015-SCI-0004.

Objective 1: test three mitigation techniques

In March 2015, three techniques were tested for their ability to mitigate the progression of yellow-band lesions on O. faveolata colonies (Fig. 1). The proposed techniques were selected based on previously published methods for treating other coral diseases, and on the current state of knowledge of disease etiology. Three replicate coral colonies were treated using each mitigation technique, and coral colonies and lesions were measured and photographed, as described in “Data Analysis” below. Because CYBD lesions spread slowly, coral colonies were revisited after four months in July 2015, and again after seven months in October 2015, and the effectiveness of each technique at reducing the rate of tissue loss was assessed.

Figure 1 Three disease-mitigation techniques that were tested on Caribbean yellow-band disease on O. faveolata.

The first method was the application of a shade cloth over the lesion (A). The second technique involved the aspiration of tissue from the lesion (B). The third technique was the chiseling of a “firebreak” around the lesion to separate it from the healthy-appearing tissue on the coral colony (C). Arrow in (B) indicates area of aspirated tissue. Arrow in (C) indicates chisel line. Photo credit: C. J. Randall.

Technique 1: shade cloth

The first technique involved the application of a shade cloth over the lesion area. This method was selected because the photosymbionts within the coral tissue are known to be affected by CYBD (Cervino et al., 2001, 2004a), and because shading has reduced the rate of tissue loss in other coral diseases (Muller & van Woesik, 2009). Reducing light and thus alleviating the potential buildup of harmful reactive oxygen species within coral tissue was hypothesized to increase immune-competence and reduce tissue loss.

Prior to deploying the shade cloths, diseased corals were identified in situ, and the lesions were measured and photographed. Custom shade cloths then were made for each lesion. The shade cloths were made of woven brown sun screen fabric 1.57 mm thick, which is rated to reduce UV light penetration by 75% (http://www.homedepot.com/p/6-ft-x-20-ft-Chocolate-Sun-Screen-Fabric-81020R/204631818). The shade cloths were cut to provide full cover of each lesion with an additional ∼5 cm of coverage of apparently healthy tissue beyond the yellow-tissue margin, when the sun was directly overhead. Lesions that were 15–20 cm long and 3–5 cm wide were targeted for treatment. Shade cloths were sewn around the edges to prevent fraying, and plastic grommets were added to the corners of each shade cloth for attachment. In situ, masonry nails were placed in each grommet and the shade cloths were installed by hammering the masonry nails into the coral so that the cloth was taut and approximately 7–10 cm above the tissue surface (Fig. 1A). Each treated colony also had at least one band that was left unshaded (absolute control). We note that masonry nail controls were not performed. In October 2015, after seven months of deployment, shade cloths were removed to evaluate the effectiveness of the treatment.

Technique 2: aspiration

Underwater aspiration of diseased tissue, followed by sealing the tissue-loss margin with modeling clay, has been used as a successful treatment for black-band disease on Orbicella spp. and other corals in the Caribbean. Hudson (2000) first developed this treatment and kindly provided his plans, which were used to custom build an underwater aspirator (Fig. 2). Briefly, the aspirator works by injecting air into a hose that runs to the surface, creating a powerful siphon. The resulting suction is then used to remove diseased tissue from the coral skeleton, which is collected in a container at the surface.

Figure 2 Underwater aspiration apparatus.

The underwater aspiration apparatus developed after Hudson (2000). Photo credit: C. J. Randall.

Our aspiration system consisted of five components (Fig. 2): (1) an air delivery system; (2) an aspiration chamber; (3) a suction hose with a stainless steel tip; (4) a discharge hose; and (5) a collection container. The air was delivered via a standard SCUBA air cylinder with a single-hose regulator, and a pressure gauge. Approximately 2 m of air hose was connected to a low-pressure port, and on the end of the air hose was the suction tube made of 1 cm diameter stainless steel tubing ∼15 cm long and angled at the tip to be used flush against the coral tissue. The suction hose was connected to the aspiration chamber and air flow into the chamber was provided by the SCUBA tank and controlled by a mechanic’s air gun. Injected air created suction through the hose, similar to an airlift pump, and collected the diseased tissue through a discharge hose. The discharge hose was a 21 m (70 ft.) standard pool hose, which was fed into a 208 L (55 gal) trash bin on a boat above the divers. The effluent then was run through an ultraviolet light sterilizer (Turbo Twist 6x; CoralLife, Franklin, WI, USA) and fed back into the surface water. The aspirator was used to remove all visible diseased tissue (band ∼15–20 m long and 3–5 cm wide) on three replicate coral colonies (Fig. 1B). Because the goal was to completely eradicate the disease from the colonies, no corals that were aspirated had untreated (absolute control) bands. Corallites were scraped with the stainless steel tube to remove as much tissue within the skeleton as possible, but small bits of tissue <1 mm2 remained in some skeletal crevices that were unreachable by the aspiration tube. No sealant was used along the aspirated margin in this study, and no procedural control for the aspirator was performed. The colonies were visually assessed at four months and seven months after aspiration.

Technique 3: chiseling a firebreak

The third technique involved the creation of a “firebreak” or trench between the lesion and the adjacent apparently-healthy tissue, using a hammer and chisel (Fig. 1C). Several studies suggest that CYBD is caused by a bacterial pathogen or consortium of pathogens that is largely isolated within the bacterial community of the diseased tissue (Cróquer et al., 2012; Closek et al., 2014; Kimes et al., 2010). If the pathogenic microorganisms causing CYBD are localized to the tissue within and around the lesion, we hypothesized that isolating the diseased tissue from the apparently healthy tissue could treat CYBD.

Three diseased-coral colonies were identified and one lesion on each colony was treated via chiseling. When present, any remaining lesions were left untreated and served as controls. A “firebreak” was established to encircle the entire lesion, with a ∼1 cm margin of healthy-appearing tissue remaining between the firebreak and the yellow-tissue margin as a “buffer” to ensure that we did not chisel directly into visibly-diseased tissue (Fig. 1C). The trench created by the chisel was approximately 1 cm deep and 1 cm wide, with the complete separation of tissue on either side of the “firebreak” (Fig. 1C).

Data analysis

To limit the introduction of physical objects into the National Park, we collected data using photographic techniques, which utilized physical features present within each colony as reference points. Photographs of all control and treated coral colonies were taken using a Canon Powershot G15 in a Canon underwater housing WP-DC48 waterproof case. Photographs were taken in Program Mode, using the Underwater Setting and auto-focus. Images were batch post-processed in Adobe Photoshop to remove blue hues and color balance the images. Each image was taken roughly 1 m away from the coral colony at approximately the same angle. Each image included a 50 cm scale bar, marked in 10 cm increments, placed parallel to the lesion. Multiple images of each colony were taken at each time point to ensure that there were comparable images across sampling periods.

To estimate the rate of tissue loss (centimeter per month) of both control and treated lesions, a “guide line” was placed on each image, parallel to each lesion, using skeletal features as a reference (Fig. 3). Three to five reference points were placed equidistant on the guide line, and the perpendicular distance from each point to the coral tissue was measured. The distance of each reference point in March 2015 was subtracted from the distance of the same reference point in October 2015, and those differences were averaged to obtain a single estimate of the rate of linear tissue over this period for each colony (Fig. 3). We note that although slight variations in distance and angle may have been introduced, the use of three to five reference points aided in the reduction of measurement variability. A one-tailed, paired t-test was used to test whether the shaded bands spread more slowly than the control bands on each of the shaded colonies (n = 3). For the aspirated and chiseled corals, the average rates of tissue-loss (i.e., the rates of spread) of the treated bands were compared with the control bands (on the shaded corals) using a one-tailed t-test. All calculations were performed in the R statistical program (R Core Team, 2017).

Figure 3 Estimating disease progression.

Method to estimate the rate of tissue loss from Caribbean yellow-band disease on O. faveolata. The “guide line” (yellow) with reference points was placed on each colony, and the perpendicular distance from each point to the coral-tissue margin was measured and the difference in distance between time 1 (A) and time 2 (B) was calculated. The distance from each reference point was averaged to estimate a colony-wide rate of tissue loss. Photo credit: C. J. Randall.

Objective 2: refine and quantify long-term success of the chiseling technique

Following the promising preliminary results from Objective 1, a more robust and long-term analysis of the chiseling method was tested. In July 2015, 19 additional corals with yellow-band disease were identified and marked with a cattle tag. At least one band was chiseled on each tagged colony and, when present, additional bands on those colonies were left untreated as controls. The trenches were further standardized, and chiseled 1 cm wide and 1 cm deep to create a substantial barrier between the lesion and the remaining apparently healthy coral tissue. In addition, two corals that had been chiseled in March 2015 were re-chiseled in spots where the apparently healthy tissue had filled in the chiseled groove and reconnected with the lesion. In October 2015, another 11 colonies were identified, tagged, and treated. In total, 11 colonies were followed for 16 months, 19 additional colonies were followed for 19 months, and the original three colonies were tracked for 23 months, for a total of 33 coral colonies treated. Of those colonies, 19 had control bands.

All chiseled corals were revisited in March 2016 and again in February 2017. During every evaluation, the condition of each chiseled lesion was classified into one of three categories: (1) “healthy”, defined as a healed tissue margin with no grossly visible signs of disease beyond the chisel line and no reconnection with diseased tissue, (2) “moderate”, defined as a healed tissue margin and no tissue loss past the chisel line, but the apparently healthy tissue reconnected with the lesion, and (3) “poor”, defined as tissue loss past the chisel line and active yellow band signs (Fig. 4). We note that the categories were applied to the treated lesion only, and not the entire colony, as many colonies had untreated lesions (controls) present in other areas. The rates of tissue loss of the treated and control bands were measured from paired images taken at the time of treatment and again in February 2017, as described above. A one-tailed, paired t-test was used to determine whether rates of tissue loss of chiseled lesions were significantly reduced compared with the control lesion on the same colonies, and a one-tailed t-test was used to compare rates of tissue loss of all control bands with all treated bands. All tissue-loss rate data were log transformed to meet the assumptions of the parametric tests. All calculations were performed in the R statistical program (R Core Team, 2017). Post hoc power analyses were undertaken to determine the power to detect an effect size of 0.8 (large effect) with an alpha error probability of 0.1 (Type I error rate) for all treatment techniques for Objectives 1 and 2, when no statistical differences were detected (G*Power 3.0.10).

Figure 4 Example images of healthy, moderate, and poor condition corals.

Images of O. faveolata colonies treated with the chiseling technique that were classified into one of three categories: (A) “healthy”; (B) “moderate”; and (C) “poor”. See text for category definitions. Arrows indicate the location of the chisel line on each colony. Photo credit: C. J. Randall.

Results

Objective 1: testing three mitigation techniques

Technique 1: shade cloth

After four months, all shade cloths remained secure and standing; the exception was one corner nail that had come loose on one cloth, which still shaded a majority of the lesion. All shade cloths, however, had become bio-fouled, effectively reducing light penetration to near 0%. Therefore, the only appreciable light reaching the coral tissue came indirectly, from ambient light around the cloth. In July of 2015, the loose nail was re-attached and all shade cloths were scrubbed with a wire brush to remove biofouling organisms. In October of 2015, after seven months the shade cloths were removed and the lesions were assessed and photographed.

All three shaded lesions continued to spread and cause tissue mortality (Figs. 5 and 6A). The lesions spread approximately 0.4 cm per month and the rate of spread was not significantly reduced under the shade cloth compared with the control bands (paired one-tailed t-test: t = −0.79, df = 4, p = 0.76). The width of discolored tissue also increased and the lesion bleached under the shade cloth (Fig. 5). We note that no negative effects resulting from the masonry nails were observed, and five months after the nail removal, the holes left by the nails had begun to heal (Fig. S1).

Figure 5 Example of the results of three mitigation techniques.

Time series photographs of O. faveolata colonies that were treated with one of three techniques to mitigate Caribbean yellow-band disease (shading [A–D], aspiration [E–H] and chiseling [I–K]). Corals were treated in March 2015 and revisited in July and October 2015. Arrows in F–H indicate the region of aspirated tissue on the colony and arrows in I–K indicate the chisel line on the colony. Photo credit: C. J. Randall.

Figure 6 Rates of tissue loss from Caribbean yellow-band disease.

Average rate of tissue loss (centimeter per month) of yellow-band disease on O. faveolata. (A) Objective 1 trials comparing aspirated, shaded, and chiseled bands with the untreated bands (“Control”). Tissue loss was measured from the initial tissue margin to the final tissue margin. (B) Objective 2 trials comparing chiseled bands with the untreated bands (“Control”). “n” indicates sample size for each treatment. “*” indicates a statistically significant differences in progression rate compared with the control (p < 0.05).

Technique 2: aspiration

All three coral colonies that were aspirated resheeted tissue over the abraded and aspirated area within four months (Fig. 5). The tissue that resheeted over the aspirated area re-grew polyps and coenosarcal tissue and was yellow, either having resheeted as diseased, or having resheeted as apparently healthy but subsequently became diseased. Resheeted tissue generally appeared visually normal in structure, although a few areas with skeletal structures that were heavily damaged during aspiration (i.e., broken corallite walls and septo-costae) had less well developed polyps. Despite some tissue regrowth, the yellow-band lesions continued to spread and cause tissue loss on the colonies. After seven months, tissue loss was comparable with control bands on the shaded colonies, advancing an average of 0.4 cm per month (one-tailed t-test: t = −0.17, df = 2.62, p = 0.56).

Technique 3: chiseling a firebreak

In July 2015, the margins of all three colonies that were chiseled had healed, and they looked apparently healthy in most areas. On two colonies, tissue regrowth over the “firebreak” reconnected the remainder of the colony with the lesion in a few small sections. However, in cases where the lesion and the main colony had not reconnected, the main colony appeared to be healthy. Yellow band was active on the lesion that was separated from the main colony, and that isolated tissue bleached and died at a higher rate (Fig. 5, see July 2015). The rate of tissue loss was not significantly reduced by the chiseling method (one-tailed t-test: t = −0.17, df = 2.62, p = 0.56; Fig. 6A), but no tissue was lost past the firebreak.

A posthoc power analysis indicated that the power to detect an effect size of 0.8 with an alpha error probability of 0.1 was 0.70 (critical t = 2.015) for the shaded corals and 0.4 (critical t = 1.440) for the aspirated and chiseled corals (G*Power 3.0.10).

Objective 2: refining and quantifying long-term success of the chiseling technique

In March 2016, 16 of the colonies treated with the chiseling technique (48%) showed signs of local recovery, defined as a healed-tissue margin and no grossly visible signs of the disease past the firebreak or around the treated area (Figs. 4 and 7; Table 1). An additional 33% of colonies showed moderate recovery, defined as a healthy and healed chisel margin but some tissue reconnection with the lesion. By February 2017, however, only the tissue around seven of the treated lesions remained apparently healthy, reducing the recovery rate to 21% (Table 1). When the treatment date was taken into account, the time series clearly indicates a steady and significant decline in the condition of the treated colonies, from more than 70% appearing healthy five months post-treatment, to less than 10% appearing healthy by 19 months post-treatment (Fig. 8).

Figure 7 Results of the chiseling technique.

Time series images of chisel-treated (A, B, C, F, G, H) and control (D, E, I, J) yellow bands on O. faveolata. Arrows indicate the chisel line on each colony. Photo credit: C. J. Randall.

Table 1 Condition of chisel-treated disease lesions.

Coral ID	Original chisel date	Condition March 2016	Condition February 2017	
CC1	March 2015	Moderate	Poor	
CC2	March 2015	Poor	Poor	
CC3	March 2015	Healthy	Moderate	
12	July 2015	Healthy	Poor	
31A	July 2015	Moderate	Moderate	
31B	July 2015	Poor	Poor	
37A	July 2015	Healthy	Healthy	
37B	July 2015	Poor	Poor	
41	July 2015	Moderate	Poor	
51	July 2015	Healthy	Moderate	
55	July 2015	Moderate	Poor	
56	July 2015	Moderate	Poor	
58	July 2015	Poor	Poor	
59	July 2015	Healthy	Poor	
60	July 2015	Healthy	Healthy	
105	July 2015	Healthy	Moderate	
107N	July 2015	Moderate	Poor	
107O	July 2015	Poor	Poor	
108	July 2015	Moderate	Moderate	
SC1	July 2015	Healthy	Moderate	
SC2	July 2015	Poor	Poor	
SC3	July 2015	Moderate	Moderate	
122	October 2015	Moderate	Healthy	
123	October 2015	Healthy	Healthy	
124	October 2015	Moderate	Poor	
126	October 2015	Healthy	Moderate	
127	October 2015	Moderate	Poor	
128	October 2015	Healthy	Poor	
129A	October 2015	Healthy	Healthy	
129B	October 2015	Healthy	Poor	
130A	October 2015	Healthy	Healthy	
130B	October 2015	Healthy	Healthy	
131	October 2015	Healthy	Moderate	
Percentage	Healthy	48.5%	21.2%	
Moderate	33.3%	27.3%	
Poor	18.2%	51.5%	
Notes:

Conditions of the Caribbean yellow-band disease (CYBD) lesions on O. faveolata that were treated via the chisel method. Corals were treated either in March, July, or October 2015. The condition of all corals was evaluated in March 2016 and again in February 2017. “Healthy” was defined as a healed tissue margin with no grossly visible signs of CYBD past the chiseled margin. “Moderate” was defined as a healed tissue margin with no tissue loss but with tissue reconnection to lesions. “Poor” was defined as the presence of grossly visible signs of CYBD with possible tissue loss.

Figure 8 Health conditions of chiseled corals through time.

Proportion of all O. faveolata colonies that were classified at each of three health conditions post chiseling, in March 2016 and February 2017. See text for definitions of each health condition. Above each bar, the month of chiseling (2015) and the sample size are indicated.

Overall, the rate of tissue loss on chiseled lesions was significantly reduced compared with control lesions, from an average of 0.31 cm mo−1 on control lesions to 0.22 cm mo−1 on treated lesions (paired, one-tailed t-test for all colonies with paired control and treatment bands: t = −2.19, df = 17, p = 0.005; Fig. 6B). This reduction was likely a consequence of similar rates of tissue loss on control and treated colonies until the firebreak was reached, where the disease was slowed or halted entirely. In addition, one colony experienced complete mortality and another colony experienced a skeletal break, therefore rates of tissue loss could only be measured on 31 colonies.

Discussion

Caribbean yellow-band disease is one of the most common coral diseases found within the Buck Island Reef National Monument (BIRNM), in St. Croix, U.S. Virgin Islands, and is significantly impacting the O. faveolata populations in the park. There is an urgent need to slow or stop the spread of CYBD on these colonies in order to preserve the O. faveolata population and increase the population’s recovery potential. Results from the experiments presented here indicate that shading and aspirating yellow-band lesions are not effective treatments. While the power to detect an effect of the shading and aspirating techniques was fairly low, the return of grossly visible signs of disease on all treated bands indicated that there is poor viability of these techniques. Chiseling a “firebreak” to isolate the diseased tissue was successful at reducing rates of tissue loss by 29%, but due to the poor long-term prognosis (Fig. 8), chiseling is not recommended for the treatment of CYBD without additional method development.

Shading corals in an effort to slow the tissue loss resulting from various diseases historically has had mixed results. Reducing photosynthetically available radiation by 40% significantly slowed the rate of tissue loss resulting from white-plague disease on Colpophyllia natans (Muller & van Woesik, 2009). However, shading corals with black-band disease increased the rate of tissue loss (Muller & van Woesik, 2011). CYBD is thought to be an infection of the endosymbiont (Symbiodinium spp.) cells (Cervino et al., 2001, 2004a, 2004b, 2008; Cróquer et al., 2012, but see Correa et al., 2009), which can eventually lead to the death of the host. If CYBD is caused by opportunistic infections of endogenous pathogenic microorganisms, it is likely that the best treatment techniques will be those that reduce stress on the host’s innate immune system, and directly treat the bacterial infection. Shading corals may partly reduce stress from high irradiance, but without concurrent treatment of the infection, it may not reduce the virulence of the pathogen(s). Indeed, shading alone did not reduce tissue loss in the present experiment, although we note that the entire colony was not shaded, and that eventually the shade cloth became biofouled, significantly reducing light penetration. The manipulative light levels may have been too low, causing light limitation. Alternatively, irradiance levels may have a negligible influence on the dynamics of CYBD.

The lack of success with the aspiration method was not surprising. Aspiration of black-band cyanobacterial mats is probably successful because the pathogenic consortium is found on the outer surface of the coral tissue, rather than embedded within the tissue, making the removal of the mat fairly easy. We hypothesize that aspirating CYBD tissue was not successful because the apparently healthy tissue regrew and reconnected with small lesion-tissue fragments that remained in the corallites that were not removable with the aspirator. That recovered tissue then began to show signs of CYBD, suggesting one or more of the following scenarios: (i) the diseased-tissue fragments remained actively infected and transmitted the pathogenic microorganisms to the re-growing tissue, or (ii) the pathogens were already present in the healthy-appearing tissue and took time to manifest macroscopically, or (iii) the pathogens may reside within the skeletal matrix and re-infect the host. Regardless of the mechanism, the aspiration technique was not successful to treat CYBD. Furthermore, aspirating diseased tissue was extremely labor intensive and required the use of a surface supply technique while live boating. Therefore, we do not recommend the use of aspiration in CYBD mitigation efforts beyond a narrow scope and without further development.

The initially promising results from the chiseling technique became discouraging as the condition of the treated colonies declined over time. However, in the few cases when the isolated diseased tissue died before reconnecting with the host coral, the coral healed and remained apparently healthy over many months. In other words, the best-case coral response occurred when the isolated, diseased tissue died rapidly and completely, preventing the apparently healthy tissue from re-connecting with the lesion. We also observed that firebreaks too close to the active area of tissue-loss allowed the disease to spread past the chisel mark easily. Additional experiments to optimize the chiseling technique are on-going and will identify the best time of year to chisel, the ideal depth and width of the “firebreak”, the ideal placement of the “firebreak” relative to the lesion margin, and the effectiveness of an epoxy application with peroxide or broad-spectrum antibiotics along the firebreak. Furthermore, the development of treatment methods that quickly and completely kill the isolated diseased tissue are being developed. We hypothesize that chiseling in late spring may be the best time of year as warm temperatures accelerate the spread of the disease (Cervino et al., 2004b) and may cause rapid loss of the diseased tissue before regrowth and reconnection can occur. However, coral growth also accelerates as waters warm (Carricart-Ganivet, 2004) potentially increasing the probability of reconnection between apparently healthy and diseased tissue. Further testing of these hypotheses, techniques, and applications are needed to determine whether the firebreak approach has any long-term viability.

In general, yellow-band disease spreads slowly, but the rate of tissue loss responds to changes in ambient conditions. Cervino et al. (2004b) documented a significant increase in the rate of tissue loss of CYBD on Orbicella spp. colonies that were inoculated with putative CYBD pathogens and maintained at 33 °C compared with controls at 20 °C, but those rates were orders of magnitude higher than those measured in the present study (measured in centimeter per hour), which are likely a result of the high concentration of pathogens in the inoculum. Also, Bruno et al. (2003) found that increased nutrient concentrations nearly doubled the rate of CYBD tissue loss on Orbicella spp., from around 0.3 cm per month to 0.6 cm per month. In the first set of trials, our rates of tissue loss averaged approximately 0.4 cm per month from July to October 2015. During the longer-term test of the chiseling technique, however, the average rate of tissue loss was closer to 0.3 cm per month. These results support previous findings that CYBD tissue loss is faster during the summer than the winter.

While the data are limited, it is noteworthy that the virulence of the CYBD pathogens at Buck Island appears to be significantly lower than it was in Puerto Rico 10–15 years ago. Weil, Cróquer & Urreiztieta (2009) measured rates of tissue loss of 0.6 cm per month in 2001–2003, but a fourfold higher rate in 2007 (2.2 cm mo−1), after the 2005 mass bleaching and disease event. These data could indicate that the virulence of the CYBD pathogens has decreased in the region since 2007. Indeed, fewer CYBD cases have been noted in the Florida Keys and in Bonaire in recent years, compared with the early 2000s (E. Peters, 2018, personal communication). Alternatively, it is possible that the resistance of host populations to the CYBD pathogens has increased, or that methodological differences between studies in measuring rates of tissue loss led to the apparent decrease. Thus, additional research is needed to formally test this hypothesis.

Conclusion

Although preventing coral diseases is the best approach, ubiquitous and endogenous pathogens that become more virulent in a warming ocean will likely continue to cause tissue loss in corals. Armed with mitigation techniques, managers may be able to work within coral restoration programs to prevent the further loss of corals after infection occurs, ultimately maintaining coral populations. While we do not support the use of the chiseling approach in its current state, additional methods that complement the chiseling technique and increase its effectiveness may become especially important as CYBD continues to affect the largest colonies that often contribute the most towards sexual reproduction within a population. Different causative agents necessitate different treatment techniques. Consequently, continuing research to pinpoint disease etiology will enable the tailoring of future treatments to the disease.

Supplemental Information

Supplemental Information 1 Effect of masonry nails.

Time series images of two representative coral colonies immediately following the removal of masonry nails from shade cloth installation (October 2015) and five months post-removal (March 2016). Masonry nail holes are indicated by arrows. Photo credit: C. J. Randall.

Click here for additional data file.

Supplemental Information 2 Raw data for disease progression rate calculations.

Click here for additional data file.

Many thanks to Nathaniel Hanna Holloway and Constance Sartor for invaluable field assistance. We also thank Monty Clark for assistance with the development of the underwater aspirator, and Justin Speaks for editorial comments.

Additional Information and Declarations

Competing Interests

Author Contributions

Field Study Permissions

Data Availability

The authors declare that they have no competing interests.

Carly J. Randall conceived and designed the experiments, performed the experiments, analyzed the data, contributed reagents/materials/analysis tools, prepared figures and/or tables, authored or reviewed drafts of the paper, approved the final draft.

Elizabeth M. Whitcher performed the experiments, approved the final draft.

Tessa Code performed the experiments, contributed reagents/materials/analysis tools, approved the final draft.

Clayton Pollock conceived and designed the experiments, contributed reagents/materials/analysis tools, approved the final draft.

Ian Lundgren conceived and designed the experiments, performed the experiments, approved the final draft.

Zandy Hillis-Starr conceived and designed the experiments, contributed reagents/materials/analysis tools, approved the final draft.

Erinn M. Muller conceived and designed the experiments, performed the experiments, contributed reagents/materials/analysis tools, authored or reviewed drafts of the paper, approved the final draft.

The following information was supplied relating to field study approvals (i.e., approving body and any reference numbers):

This project was conducted under permit BUIS-2015-SCI-0004 issued by the National Park Service (USA).

The following information was supplied regarding data availability:

The raw data of all replicate measurements of disease progression on each colony are included in the Supplemental File.

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
