# Peer review of "Testing methods to mitigate Caribbean yellow-band disease on Orbicella faveolata"

_PeerJ, doi:10.7717/peerj.4800_

## Round 0.1 · original submission · Major Revisions

The reviewers found this submission of great interest and all consider that this would be a sound contribution to the field. However before the manuscript can be considered for publication the authors require to address the constructive comments and suggestions provided by the three reviewers. It is important the authors clarify concerns about experimental controls and make sure they use the right disease/pathology terminology.

Reviewer 1 ·

Basic reporting

The manuscript is generally well written with appropriate, professional use of English throughout. I recommend the author consider restructuring the introduction. The opening paragraph describes Caribbean yellow-band disease distribution and its history in five sentences. The next paragraph discusses generally, coral disease outbreaks followed by a paragraph that outlines previous treatment attempts for various types of disease, then back to Caribbean yellow-band disease for the next three paragraphs. This structure is a bit disjoint and I would recommend moving the opening paragraph between the second and third paragraphs (after line 85). I believe this would provide a more logical flow from general to more specific. There likely will need to be additional text to create a bridge from the general treatments to more explicitly connect the concept of treatments to CYBD.

The discussion summarizes the findings with additional commentary that provides explanations/interpretations for the treatment outcomes or suggested improvement.

Line 100-101 – The logic behind the statement that indicates CYBD is caused with weakened host immunity seems weak. Lines 87-88 describe CYBD as primarily a disease of the symbiotic dinoflagellates. It doesn’t necessarily follow that it is the host’s defenses that are compromised. There are numerous references that describe various defense systems in algae, including dinoflagellates, thus it is just as likely that it is dinoflagellate's defense system that is compromised as that of the host. Recommend reconsidering this statement since it is speculation and modify.

Line 349 – please clarify whether the epoxy application was intended to suggest a mixture of peroxide and antibiotics or should it read ‘peroxide or broad spectrum antibiotics…’?

Lines 376-378 – Recommend that the authors consider other alternatives to just a decrease in virulence when comparing Puerto Rico to the Buck Island populations. For example: Were the measurement methods comparable? Were the genotypes of Orbicella sp. comparable?

Figure 4 – The image in the shaded x July 15 appears to be a different magnification than the others in the shaded series of photos. For other images, particularly in the chiseled series add small arrows to depict areas described in the text to assist reader in orienting to the particular area of interest being described. I would suggest also for the authors to investigate whether color correcting to remove some of the blue hue in these photos could help increase the contrast and clarity for the reader, since this panel of photos is relatively small. The methods do not describe the photography, but it appears that the images could have been improved with proper lighting, for these photos as well as others in the manuscript.

Figure 6 – Similar comments as for Figure 4 regarding color correction. Also arrows or other notations that correspond to the text description of these images would be helpful.

Experimental design

Overall the experimental design is sound, is sufficiently replicated and demonstrates three different approaches to treating Caribbean yellow-band disease and the treatment outcomes. It is also worth noting that the extended observation time was critical to the final conclusion regarding the efficacy of the chiseling method. Although a fully successful treatment was not discovered, means of treating priority, high valued coral in the field is critical for coral conservation given the current state of disease in many coral populations, which exceeds the normal carrying capacity of disease in a population and this work moves us a step forward in identifying treatments that work.

Most of the methods are described in sufficient detail to evaluate their soundness and allow replication. However, this falls short in the description of the digital photography and details for the image analysis and software used. Given the photos provided in the various figures it is unclear how measurements were made to sub-centimeter accuracy. Though the text indicates photos were taken from the same angle, the scale used seems quite coarse and often at various angles in the examples provided. The distance from the subject is also important in a repeated measures design. A more thorough explanation by the authors on how they achieved the precision of the reported rates of lesion movement is important to include in the manuscript as well as any caveats. It is well known that even slight differences in angles between camera and lesion, placement of the scale or ruler and distance from the image can generate significant differences in the resulting measurements. The authors are referred to Chapter 35 of the book Diseases of Coral by Moses and Hallock. This chapter provides details of the digital photography methods and image analysis for assessing wound healing in coral. Also suggested is that the authors review the method of digital planimetry (e.g., Plos One 10(8): e0134622 and other references). For precise measurements there are specific criteria that need to be followed, it is not clear that these were used here.

Validity of the findings

Although there is concern regarding the precision of the measurements reported, the overall findings and conclusions appear sound. There is sufficient replication of the treatments in the design and statistical analyses are sound. I believe the suggestions and recommendations detailed in other sections of this review can provide improvements to the manuscript.

Additional comments

1. Lines 46-59 – the sentence beginning on line 57 seems redundant with a previous sentence beginning on line 52. Recommend rewording sentence on line 57 to: While it is seldom reported prior to the 2005 Caribbean coral bleaching event, CYBD is now common on many Caribbean reefs.
2. Line 70 – Suggest using a more specific description of the ‘modeling clay’ to avoid confusion. I believe it is a specific artist clay with defined characteristics (Roma plastilina #2) that was used in the reference and subsequently recommended by the FL Keys National Marine Sanctuary for backfilling divots or bare skeleton resulting from sampling or treatments.
3. Line 73 – suggest breaking the sentence beginning on line 72 after ‘mixed success’. And begin next sentence with “These include; …”
4. Line 91 – suggest breaking sentence after the citation list and begin the next sentence “Further, Koch’s postulates…”.
5. Line 92 – suggest revising sentence to reduce wordiness to “However, studies clearly show that the bacterial communities…”

Reviewer 2 ·

Basic reporting

This is an interesting paper because it explores one of the most neglected fields on coral disease research which is the development of treatments to mitigate the effects of disease on their host, to cure or to expand the life spam of infected hosts. It is also valuable because the model-host-pathogen selected is without hesitation among the most worrisome problems on Caribbean reefs nowadays. The paper is well-written, literature review is pertinent, hypotheses are clear as well as results. There are some minor statements which I consider to be wrong (e.g. CYBD lessons extend perpetually). While this may actually happens, it is not always like that, some times the disease arrest, progression slows down and even stops for years and in some cases the disease reactivates. Yet, I have seen colonies that might fully recover. This pattern is actually an unexplored field perhaps determined by a combination of environmental-genetic factors affecting host-pathogen interactions, resistance, susceptibility and virulence. I think this needs to be further discussed in the manuscript. Like in many host-parasite systems, there are some host that can cope with the infection better than others, and that also applies to treatments because not all hosts respond in the same way to them. Question is why?

There are other interesting findings of CYBD that are missing in the introduction that can be used for this paper. For example, Guerra et al. (2014) found that the gross (relative) chemical composition of CYBD tissues are different compared to healthy tissues.

In Material and Methods, there is an assumption for selecting the treatment of shading: "shading reduces light and the potential built up of ROS which should increase immune competence" In my opinion this assumption is not quite well supported for CYBD. There are a couple of papers showing that immunologic responses in CYBD tissues are compromised (e.g. Mylardz et al. 2009, Montilla et al. 2016). So I do not think, that shading will improve immunological responses in this case, cause CYBD tissues have been shown to be compromised already.

As for experimental design. There are some points that in my opinion deserve some discussion and in other cases some relevant considerations that need to be clarified in order to better judge the validity of the results presented here. Briefly:

(1) I am happy with the application of the treatments but I am not clear with the controls that were used. Authors talked about controls, but I am not sure if these are absolute controls (i.e., diseases colonies without the treatment, no shading for instance) or procedural controls (e.g. a set of colonies with a frame similar to the one used for shading but allowing light to penetrate). For the aspiration treatment, the control would be no aspiration (untouched corals) and the procedural control the same mechanical treatment but without removing the tissue completely (i.e., vacuum off). As for the fire wall, the procedural control would be scratching the tissues without creating a deep trench in the tissue.

(2) According to the results, authors did not find an effect. Given the low umber of replicates used (3 colonies), I think it is necessary to estimate the achieved statistical power. In other words, it is imperative to convince the reader that authors did not found a false negative result (i.e., retain the null hypothesis when it is wrong).


Discussion and conclusions: the literature review is Ok, the discussion is focused on author's results. However, I think it is necessary to better explain the nature of controls used in the experiment (i.e., absolute or procedural) to better judge the validity of these findings.

Experimental design

The paper is original and interesting. The questions are clearly stated and well supported from theoretical backgrounds, ethical standards are fulfilled, the technical quality of the paper, however, needs to be revised for more detail on methods and experimental design used needs to be presented (see below).

Please explain the use of controls in the experiment. I am not sure if these are absolute controls (i.e., diseases colonies without the treatment, no shading for instance) or procedural controls (e.g. a set of colonies with a frame similar to the one used for shading but allowing light to penetrate). For the aspiration treatment, the control would be no aspiration (untouched corals) and the procedural control the same mechanical treatment but without removing the the tissue (i.e., vacuum off). As for the fire wall, the procedural control would be scratching the tissues without creating a deep trench in the tissue.

Validity of the findings

The results are novel. Replication seem to be low, i.e., only three colonies per treatment. While this is not a problem per se, I believe that power analysis must be presented. For instance, what is the maximum achieved power given a size effect and the actual total variance? What would be the sample size that was required to achieved the power desired? It is important to show the readers that there no effects for two out of three treatments and this is not a problem of a huge type 2 error due to low replication.

Conclusions will be validated if authors convince me why the lack of procedural controls like the ones used, for instance, in classic exclusion experiments whereby treatments are: cage (complete exclusion), open treatments (no cage or exclusion absolute control) and cages without cap (procedural control for the device or artifact used in the experiment).

Additional comments

I think this a great idea and a great paper. The literature review is very good, the way hypotheses and models supporting the hypotheses are presented is very nice. While I think sample size for each treatment is low, i celebrate the temporal component of the experiment. It is very nice to see that colonies were monitored for long time. I think this sort of papers are needed and represent an important contribution for coral disease research.

My only concern is the use of controls, I think, there is a lot of manipulation in this experiment, therefore artifact controls (not only absolute controls) were needed.

I would like to see and looking forward to listed author's arguments before I decide whether or not this paper can be accepted for publication.

·

Basic reporting

This is an excellent manuscript in terms of writing and editing, making it a pleasure to read! Background research on the problem studied is solid and focused; photos of the field experiment and detailed illustrations are included, and raw data are shared. However, coral scientists continue to use terminology that is inappropriate or incorrect to describe what is happening as coral tissue disappears from the skeleton in diseases. Terms and descriptions from pathology are needed. For example, the definition of "disease progression" is “The worsening of a disease over time” https://www.bing.com/search?q=disease+progression+definition&form=EDGHPT&qs=AS&cvid=bef1590c7876433c9aa6a5747bff3eb1&cc=US&setlang=en-US. “Progression” and “advancement” have been used in papers on coral diseases when explaining the continuing loss of tissue from the skeleton, as seen in several diseases given various field-identifying names, such as white plague, white-band disease, and Caribbean yellow-band disease. However, the disease is not worsening in severity or intensity over time, just its effects, i.e., the amount of coral tissue killed at the colony margin (not “front”) has increased. "Front" is not being used correctly in the phrase “disease front.” “Front” refers to an orientation, as in what someone or something is facing, or faces forward or in a particular direction, or an area along the edge of something, not the edge itself. That should be changed to “tissue-loss margin.” In multiple places "disease [or diseased] lesion" is used. A lesion is any area of abnormal tissue and it is abnormal when it has a functional impairment of any kind, therefore it is diseased, by definition. Also, lesions do not “advance,” as in “move,” but enlarge as more cells die, lyse, and disappear from the skeleton. Finally, the term "virulence" is often used with "disease," when virulence, which is a pathogen or microbe's ability to damage or infect a host, should refer to the pathogen (any virus, microorganism or factor, biotic or abiotic, that causes disease). Detailed edits on necessary wording changes are provided in the general comments section.

Experimental design

This manuscript reports on well-designed experiments to test hypotheses of methods that might be used to control tissue loss in CYBD-affected corals. The literature review supports the gap in knowledge and the experiments could probably be replicated based on the information provided. They were performed with high technical and ethical standards. The field permit is identified in line 130.

Technique 1: Number of lesions on number of colonies treated is not provided in text (is in results figure, but could be stated in text, too).

Technique 2: Supplemental Figure 1, why not put this with the main paper? In middle text box, “syphon” should be spelled “siphon” (the latter is the more common or usual spelling, at least in the US, also change it elsewhere, as in Line 167)

Line 225: not exactly sure what is meant by “had reconnected with the disease lesion.” In Figure 3b, it appears that the tissue remaining outside of the chiseled groove has regained normal color and what was apparently healthy tissue at the time of the chiseling has now filled in that groove to reconnect with the formerly diseased tissue. So, may be best to write “had filled in the chiseled groove and reconnected to the formerly diseased tissue”?

Validity of the findings

The findings are consistent with the methods used and the resulting data obtained. The discussion is well done, except for including a paper that is not relevant. The supporting literature is otherwise appropriate and the conclusions are valid, with good suggestions for future research and application of tissue loss control techniques for diseased corals. The conclusions are well stated, cover the original research question, and limited in scope to supporting the results.

Additional comments

Line 26: change “lesions advance slowly” to “lesions enlarge slowly”; not all cases result in complete loss of tissue from a colony, so change “eventually leading to” to “often resulting in”

Lines 27-28: change “The slow advancement of CYBD, however,” to “The slow loss of tissue in CYBD, however,”

Line 30: change “mitigate disease progression” to “mitigate the disease” or “minimize disease impact” (last preferred, if leaving “to mitigate CYBD” on line 9)

Line 32: change “progression” after CYBD to “tissue loss”

Line 33: change “rate of disease progression” to “rate and amount of tissue lost”

Line 37: change “slows disease progression” to “slows tissue loss”

Lines 42-43: change “slow and variable progression” to “slow but variable rate and pattern of tissue loss in this disease.”

Line 76: change “white-sign” to “tissue-loss”

Line 93: change “on diseased lesions” to “on and in diseased coral tissues” AND change “on healthy colonies” to “on and in apparently healthy coral tissues”

Line 106: change “the lesion front” to “the tissue loss margin”

Line 125: change “thick” to “wide”

Lines 126-128: confusing description of the lesions here, suggest changing to “A gradient of tissue discoloration is present, with apparently normally thick and pigmented tissue progressively thinning and becoming paler yellow at the dying margin, which is usually smooth with an indistinct edge. The algae biofouling community that colonizes the denuded skeleton is visible adjacent to this edge, i.e., areas of stark white recently denuded skeleton are rarely seen.”

Lines 128-129: change “progresses slowly, either” to “proceeds slowly, as the tissue-loss lesion expands, either”

Line 137: change “progresses slowly” to “tissue loss proceeds slowly”

Line 139: change “disease progression” to “the rate of tissue loss”

Line 143: change “progression” to “tissue loss”

Line 146: change “disease progression” to “tissue loss”

Line 152: change “of coverage of healthy tissue beyond the disease front” to “coverage of apparently healthy tissue adjacent to the thinning yellow tissue margin”

Lines 153 and 374, 375: Need en dashes instead of hyphens between the numbers of cm to mean “to” in the range of values

Line 162: change “sealing the diseased-tissue front” to “sealing the tissue loss margin”

Line 190-191: change “diseased lesion” to “lesion”, insert “apparently” between “adjacent” and “tissue” and delete “that did not show visible signs of disease.

Line 191: “a bacterial pathogen” but in some papers the disease might be caused by multiple bacteria species?

Line 193: change “If CYBD is” to “If the pathogenic microorganisms causing CYBD are”

Line 199: change “and the disease front” to “and the yellow tissue margin”

Line 204: change “of progression of both control lesions and treated lesions” to “of tissue loss from control lesions and treated lesions”

Line 210: change “estimate of linear disease” to “estimate of the rate of linear tissue loss”

Line 211: change “progression” to “during this time (disease progression)”

Line 212: change “shaded bands progressed more slowly than” to “tissue loss on the shaded parts of bands was slower than”

Line 213: change “progression” to “tissue-loss”

Line 223: change “barrier between the diseased lesion and the remaining coral tissue” to “barrier between the lesion [OR the diseased tissue] and the remaining apparently healthy coral tissue”

Line 224: add “apparently” before “healthy”

Line 225: not exactly sure what is meant by “had reconnected with the disease lesion.” In Figure 3b, it appears that the tissue remaining outside of the chiseled groove has regained normal color and what was apparently healthy tissue at the time of the chiseling has now filled in that groove to reconnect with the formerly diseased tissue. So, may be best to write “had filled in the chiseled groove and reconnected to the formerly diseased tissue”?

Line 230: “condition” would be better term than “health,” since that should be determined by multiple parameters, not just gross observations

Line 231: here ‘healthy’ has been defined as being based on gross observations, so this is fine; change “no macroscopic” to “no grossly visible” (also on line 281)

Line 233: change “disease progression” to “tissue loss”

Lines 233-234: “the diseased lesion” is unclear. Would the tissue have reconnected to diseased tissue, was this seen (not apparent in Figure 3b)? Would changing this to “the formerly diseased tissue” or “the now apparently healthy tissue” be better here?

Line 236: delete “disease” before “lesions”

Lines 237–241: Change “progression rates” to “tissue-loss rates” or “tissue loss-rate data”

Line 255: change “to progress” to “to enlarge”

Line 256: change “The disease front advanced….and the rate of progression was not significantly reduced” to “The tissue loss rate was approximately 0.4 cm per month and was not significantly reduced….”

Line 258: change “diseased lesions also appeared to widen and bleach” to “width of the discolored tissue at the lesion margin increased and bleached…”

Lines 263-266: A very interesting observation! Did the tissue include polyps and coenenchyme, and were these visually normal in structure and behavior, if not also color? Perhaps the only zooxanthellae that were available to repopulate the tissue was the clade reported to be resistant to the pathogen(s) (whatever it is), which is why the discoloration is translucent yellowish in this disease. Need to describe in more detail what was observed, to qualify the terminology. Change to “having resheeted as diseased or having resheeted as apparently healthy but subsequently became diseased.” End of line 264: change “advance and cause tissue loss on” to “enlarge with additional tissue loss on”; line 266: change “advancing” to “continuing at”

Lines 271-272: change “disease lesion” to “lesion”; change “main” to “remainder of the colony”

Lines 274-277: change “with high rates of progression” to “at a higher rate”; change “The progression of the disease lesion” to “The enlargement of the lesion”; and “had progressed” to “had developed”

Line 284: delete “diseased” before “lesion”

Line 285: insert “apparently” before “healthy”

Line 287: change “health” to “condition”; “over” to “more than”

Line 290: change “rate of disease progression” to “rate of tissue loss”

Line 294: change “progression” to “tissue-loss”

Line 295: delete “progression” after “disease” (tissue loss is a key feature of the disease)

Line 297: change “progression rates” to “tissue-loss rates”

Line 308: change “disease progression rates” to “tissue-loss rates”

Line 308: change “outlook” to “prognosis”

Line 310: change “progression of various” to “tissue loss resulting from”

Line 312: change “progression” to “rate of tissue loss”

Line 313: change “rate of disease progression” to “rate of tissue loss” or “rate of lesion enlargement”

Line 317: change “infections of endogenous pathogens” to “infections of endogenous pathogenic microorganisms” [pathogens can also be non-infectious]

Line 320: change “of the disease” to “of the pathogens”

Line 321” change “progression” to “tissue loss”

Line 330: insert “apparently” before “healthy”

Line 330: change “disease” in both places to “pathogenic microorganisms” (and “was” to “were” after the second one)

Line 341: change “health” to “condition”

Line 343: insert “apparently” before “healthy”

Line 345: delete “diseased” before “lesion” OR change “lesion” to “tissue”

Line 346: change “lesion” to “tissue loss” AND “progress” to “proceed”

Line 349: change “disease front” to “tissue loss margin”

Line 350: Note that the coral disease community is not advocating use of antibiotics in the field, perhaps this should not be mentioned or state that this will only be tried in controlled laboratory conditions?

Line 353: change “lesion” to “diseased”

Line 355: insert “apparently” before “healthy”

Line 358: change to “yellow-band disease slowly affects tissue, but the rate of tissue loss responds…”

Line 359-362: The “yellow-band disease” described in the Korrubel and Riegl paper was not at all similar to what is now known as CYBD, not only in location and species affected, but also in the description of the lesion. In personal communication with B. Riegl, this reviewer learned that the “yellow band” was more like the fine filaments seen in black-band disease (BBD), but brightly yellow, and that yellow particles remained on the recently denuded skeleton after the tissue disappeared. A large pile of mined elemental sulfur was onshore near this site. Because the microbial consortium in BBD includes all the members of the sulfur cycle, including Beggiatoa spp. that excrete particles of elemental sulfur on the surface of their filamentous cells, what might have happened is that sulfur from the pile had run off or blown into the seawater there and black-band microbes had been metabolizing the excess sulfur. Unfortunately, no other studies were done on these affected corals, including microbiological or histological evaluations, so we cannot be sure. However, this reviewer suggests deleting this reference, because it is not at all related to CYBD, and that rate of tissue loss was similar to what occurs in BBD, another clue to its identification.

Line 362: delete “also”

Line 363: change “progression” to “tissue loss”

Line 368: change“progression” to “tissue loss”

Lines 369 and 371: change “progression” to “tissue-loss”

Line 372: change “CYBD progresses” to “CYBD tissue loss is”

Line 373: “virulence of CYBD” is not correct. In pathology, a disease is not virulent, but a pathogen (abiotic or biotic) has a quality known as virulence, which is a pathogen or microbe's ability to damage or infect a host (“virulence of a disease” may be found in general dictionaries, but this is not used in pathology). This could be changed to “the virulence of the CYBD pathogen(s)”

Line 375: change “progression” to “tissue-loss”

Line 377: change “virulence of CYBD” to “the virulence of the CYBD pathogen(s)”

Line 379: change “coral-disease infections” to “coral diseases” [pathogen may not be biotic]

Line 382: change “infection occurs” to “tissue loss starts”

Excellent ending!

References – some consistency issues in capitalization and formatting

Line 425: “Vibrio” should be “vibrio”

Line 439: fix to “Current Microbiology”

Line 442: for the record, disease names are not capitalized, but yes, this was unfortunately done in this paper title.

Lines 444, 486, 487: book chapters, should follow formatting of reference on Line 410

Line 460: suggest deleting from text and here

Line 477: italicize journal title and volume number

Line 480: fix to “Coral Reefs”

Line 495: italicize “Acropora”

Figure 1 legend begins with “Three disease mitigation techniques.” Then next paragraph says “Three disease-mitigation methods…” Need to put hyphen between “disease” and “mitigation” at beginning to be consistent. The middle 3 lines contain the words “disease lesion” but that only needs to be “lesion”

---

## Round 0.2 · Minor Revisions

All reviewers have provided additional comments that will help to improve the manuscript. Send a revised version with each of this suggestions/corrections addressed.

Reviewer 1 ·

Basic reporting

I have conducted a second review of this manuscript after revisions. I find all of the revisions acceptable and the manuscript suitable for publication with the following exceptions:
Line 75 “application of chlorine embedded within epoxy to mechanically and biologically block” It is unclear the intent of ‘mechanically and biologically’ but chlorine is a chemical and thus is not a biological agent for blocking. Rephrase to clarify meaning of biologically or change to chemically or other relevant term.

Line 79 – the reference should be Gignoux-Wolfsohn et al. 2012. Unless specified by PeerJ it is conventional to use et al. if there are more than two authors.

Line 128 – ‘apparently normally thick…’ should be rephrased. Suggest ‘….with apparently normal thickness and pigmented….. OR ‘apparently normal, thick and pigmented tissue’ depending on the intent.

Line 129 – ‘which is usually smooth with an indistinct edge’ Suggest clarifying, what is smooth and how is it then an edge indistinct?

Line 129 – correct spelling of ‘biofouling’

Line 131-132 – though I think I understand “centrally located lesions vs peripherally located lesions”, I’m not sure a reader unfamiliar with CYBD would know what this means. Suggest either providing a figure of the two or improving the description.

Lines 472-474 – This is not the most appropriate reference (in addition to being a single author and the text reflects Fisher et al.) Use instead Marine Ecology Progress Series (2007) 339:61-71.

I believe these are minor and could clarify the manuscript.
Two of the above are corrections that need to be made.

Experimental design

meets standards

Validity of the findings

meets criteria

Reviewer 2 ·

Basic reporting

Overall I am happy to see that authors undertook major issues of experimental design and data analysis. I am quite happy with their review.

There is however minor issues that needs to be considered before the paper goes into print. First, the paper now acknowledge the lack of procedural controls (i.e., controls for the artifacts used during the experiment), and I think that is honest and great. Authors mentioned that they did not notice any effect of these artifacts (e.g. nails, shading devices, etc) and I believe them. Yet, I think that a paragraph or at least a sentencein the discussion arguing about the potential effects of ignoring these sort of controls on the results is needed. I strongly advice authors to include procedural controls in their future experiments.

Secondly, there is a conceptual problems (minor one) with the power analysis they performed. Power analysis only proceeds when you don't find significance in your main test. Authors included power analysis in cases where statistical significance was found. They need to delete this from the manuscript.

One can calculate the rate of a false positive for a test that becomes significant, if it is desired, but this is not a power analysis.

I will be happy to endorse the publication once these minor changes are incorporated.

Experimental design

The problem of experimental design cannot be solved unless the experiment is repeated, but at least the authors acknowledge the problem. I think that is the way to go. However, I think that a sentence in the discussion explaining how this may hamper the conclusions is needed. As a reader, I believe in the "qualitative-observational" argument that says "we do not observe any effect" but I also want to read that if artifacts used in the experiments had an effect, how these could have had hampered the results and conclusions.

Validity of the findings

Again, anyone could say that the lack of procedural controls invalidate the results. In this case I don't; but authors need to go a bit over this issue in the discussion.

Additional comments

I congratulate authors for this paper, I do think is a great contribution. I strongly recommend to delete power analysis in cases where statistical significance is found. I celebrate how authors undertook my comments, I am quite happy with that. Yet, I think a sentence in the discussion is need it to argue how the lack of procedural controls would hamper (or not?) the results.

If these minor changes are included, I will be happy to endorse this paper for publication.

·

Basic reporting

I approve of the revisions made to the manuscript, it is much improved.

Experimental design

Additional details and explanations provided in the revision have answered the questions posed by reviewers.

Validity of the findings

The data validity remains sound and additional details on the analyses continue to support the findings.

Additional comments

The revisions made are acceptable and the paper is now in excellent shape! Just a couple of minor things:

Use of "spread" vs. "enlargement" of lesions: Seeing that the co-authors used the former, I looked at papers online to see if there is a preference or particular definition of these terms for pathology. It appears that "spread" is appropriate, sometimes these terms are used interchangeably, or to mean different things (as in a focally, size-limited type of lesion appearing in an increasing area is spreading, whereas the increase in size of a singe lesion is enlarging). So, this reviewer would prefer if distinction could be made for the lesions studied here, but understands that may not be possible.

Line 382: change "progression" to "spread"?
Line 385: add "d" at the end of "disease"
Line 399: break into a new paragraph at "While the...", because it is rather long.

Line 404: the authors note the prevalence of CYBD has decreased in some areas with time. This reviewer notes that fewer cases were seen on Looe Key Reefs in 2011 and it was absent on Bonaire reefs she visited in 2017, whereas it had been rampant there in the early 2000s, but surviving Ofav colonies had continued to grow, building skeleton and live tissue around suspected former lesions.

Line 61: the sentence change made here is good; this reviewer has a photo of an Ofav colony with suspect CYBD taken in 1979 on Grand Cayman reefs and had heard that others were reviewing their photo collections to look for it. (Thought the lesions might have been due to the clionid sponge appearing in the exposed skeleton, but of course, this was before we would think much about coral diseases!)

---

## Round 0.3 · accepted · Accept

The authors have addressed satisfactorily the comments and suggestions by the reviewers.

#